



# Saharan and Arabian dust optical properties registered by sun photometry during A-LIFE field experiment in Cyprus

David Mateos[1], Carlos Toledano[1], Abel Calle[1], Roberto Román[1], Marcos Herreras-Giralda[2], Ramiro González[1], Sara Herrero-Anta[1], Daniel González-Fernández[1], Celia Herrero-del Barrio[1], Argyro Nisantzi[3,4], Silke Groβ[5], Victoria E. Cachorro[1], Ángel M. de Frutos[1], Bernadett Weinzierl[6]

[1]Grupo de Óptica Atmosférica, Laboratory for Disruptive Interdisciplinary Science (LaDIS), Universidad de Valladolid, Paseo Belén 7, Valladolid, Spain
2 GRASP SAS, Remote Sensing Developments, 59260 Lezennes, France
3 Eratosthenes Centre of Excellence, Limassol, 3012, Cyprus
4 Department of Civil Engineering and Geomatic, Cyprus University of Technology, Limassol, 3036, Cyprus
5 Deutsches Zentrum für Luft- und Raumfahrt (DLR), Germany
6 Universität Wien, Aerosol Physics and Environmental Physics, Austria

*Correspondence to*: David Mateos (mateos@goa.uva.es)

**Abstract.**

The A-LIFE (Absorbing aerosol layers in a changing climate: aging, lifetime, and dynamics) field experiment in Cyprus (April 2017) employed a wide range of ground-based and airborne instruments, including passive/active remote sensing and in-situ techniques. This study presents the columnar records obtained by sun photometry. Two sun/sky/lunar photometers, belonging to AERONET network, were strategically placed at two different sites: Pafos and Limassol, 40 km apart. Aerosol optical and microphysical properties derived from direct sun and sky radiance measurements are analysed to determine an inventory of aerosol event days during the whole experiment, with mineral dust being predominant. Obtained Ångström exponent values in the near-infrared range (0.5 for Saharan dust and 0.34 for Arabian dust) can served as a classification criterion. Dust sources are the key point for a well understanding of the size distribution and absorption power. Saharan dust exhibited smaller and less absorbing particles than Arabian dust. The columnar volume efficiency factor (linear fit between aerosol optical depth and total volume concentration) was proved as a reliable proxy for the identification of dust origin since Arabian and Saharan dusts exhibit different slopes: 1.28 and 1.68 $\mu m^2/\mu m^3$, respectively. Mixtures of mineral dust were mainly dominated by Arabian dust, while mixtures of fine and coarse aerosols showed no clear prevalence of dust origin. No significant presence of black carbon-rich aerosols was detected in the atmospheric column, as absorption Ångström exponent values ranged from 1.6 to 3 across aerosol types identified in the inventory of A-LIFE experiment.



## 1 Introduction

Wind-blown dust is one of the key aerosol types in the Earth's atmosphere, with an estimated global emission flux of approximately 5000 Tg yr$^{-1}$ and with geometric diameter up to 20 μm (Kok et al. 2021). Mineral dust significantly affects the climate by interacting with solar radiation (Liao and Seinfeld, 1998), atmospheric and surface temperature, cloud properties

and processes (Ansmann et al., 2019) and air quality and human health (Knippertz and Stuut, 2014). The primary sources of dust are in arid regions, mainly deserts, where natural emissions exhibit marked seasonal cycles (e.g., Kubilay et al., 2003; Alizadeh Choobari et al., 2014; Prospero et al., 2014).

Sahara Desert and Arabian Peninsula are two of the most important sources accounting more than the half of the global dust emission (Tegen and Schepanski, 2009; Notaro et al., 2013; Klingmüller et al., 2016). Once in the atmosphere, dust particles

can be carried across vast distances being one of the main aerosol types even in regions distant from their sources. Dust transport pathways are generally seasonal dependent. For instance, connections between Saharan desert and Caribbean Basin are favored in summer and dust storms in the northern/southern Arabian Peninsula are more frequent in summer/spring (e.g.,; Prospero and Mayol-Bracero, 2013; Alizadeh Choobari et al., 2014; Velasco-Merino et al., 2018; Euphrasie-Clotilde et al., 2020; 2025).

The dust chemical composition varies from the origin source (Castellanos et al., 2024, and references there in). Su and Toon (2011) found differences in the shape of particle size distribution between Asian and African dust. Larger and less absorbing particles are expected in Asian dust. Variations in iron or hematite contents can influence optical properties (e.g., Myhre and Stordal, 2001; Claquin et al., 1998; Di Biagio et al., 2019). Dubovik et al. (2002) reported differences in size distribution in two Arabian and one African site. Different dust geographic origins are obviously linked to different columnar properties.

These differences in composition and size lead to different absorption properties. Absorption properties by dust aerosols contribute to an increased heating rate of the aerosol layer, but mixtures of dust with other aerosol types can modify its absorption power (e.g., Alizadeh Choobari et al., 2014). There is a negative correlation between absorbing aerosols and cloud development since mineral dust embedded within clouds absorb radiation, resulting in a decrease in relative humidity and in cloud cover (e.g., Ackerman et al., 2000; Huang et al. 2006)


In this framework, the "absorbing aerosol layers in a changing climate: aging, lifetime and dynamics" (A-LIFE) project aims to investigate the properties of absorbing aerosols, focusing on mineral dust and its mixtures with black carbon and how they are distributed throughout the atmospheric column. A major part of the project was a field campaign conducted in April 2017 in Cyprus, since this island is a hot spot regarding atmospheric aerosols because of its proximity to Saharan Desert, Arabian

Peninsula and European and Asian Continent. This field experiment was a big opportunity to analyze data from collocated highly-sophisticated in-situ and remote sensing instrumentation, meteorological sensors, and series of instrumentation onboard Falcon research aircraft of the German Aerospace Center DLR (see https://www.a-life.at for further details). Our contribution to this project oversees columnar aerosol properties collected from CE318-T (*Cimel Electronique*) sun-sky-moon photometers in the AERosol RObotic NETwork (AERONET, Holben et al., 1998) in two different Cypriot locations: Pafos Airport (where

the in-situ station was placed), and Limassol University (CUT-TEPAK site, close to ground-based multi-wavelength lidars BERTHA and POLIS, see Groß et al., 2024). The aim of this study is, therefore, to analyze in detail aerosol columnar properties during A-LIFE experiment. An inventory based on columnar records is presented here. To our knowledge, it is the first time that Saharan dust, Arabian dust, and their mixtures are jointly analyzed focusing on the different absorption properties obtained by sun photometry.



## 2 Aerosol Columnar Measurements during A-LIFE

The aerosol columnar measurements presented in this study were taken at different sites belonging to AERONET network, both in Cyprus Island. The main site used in this study is CUT-TEPAK, in Limassol (34.675ºN, 33.043ºE, 22 m a.s.l) where several collocated remote sensing instruments were installed during the A-LIFE field experiment such as the mentioned sun-

sky-lunar-photometer CIMEL, SSARA sun-sky radiometer (Toledano et al., 2009), POLIS lidar (Groß et al., 2015, 2016) and the BERTHA lidar (Haarig et al., 2017). Meteorological radiosondes were also available during the campaign. As a complementary site, a second CIMEL photometer was installed at Pafos airport (34.711ºN, 32.483ºE, 10 m a.s.l.) only during the field experiment and 44 km away from Limassol city. This site was selected because it was close to in-situ instrumentation, including samplers and optical instrumentation for derivation of dust optical, chemical, and micro-physical properties at ground

level (Kristensen et al., 2016; Kandler et al., 2018). A description of the instruments available during A-LIFE field experiment is provided by Groß et al. (2024) and Teri et al. (2025).

The standard instrument used in AERONET is the Cimel CE318 radiometer, which performs direct sun measurements across several wavelengths in the spectral range of 340–1020 nm. Additionally, the instrument measures sky radiance in the solar almucantar and hybrid configurations at wavelengths of 440, 670, 870, and 1020 nm (see Holben et al., 1998).

Direct sun observations are utilized to derive the spectral aerosol optical depth (AOD) and the corresponding Ångström exponent (AE) by fitting AOD between 440 and 870 nm. The database used in this study presents the maximum quality level assured by AERONET (level 2.0, version 3), presenting an AOD uncertainty about 0.02, larger for shorter wavelengths (Giles et al., 2019).

The sky radiances at four wavelengths (440, 675, 870, and 1020 nm), combined with the AOD, are employed to retrieve a set

of aerosol optical and microphysical properties via inversion methods (Dubovik et al., 2006). These properties include particle size distribution, complex refractive index, single scattering albedo (SSA), phase function, absorption AOD, etc. To analyze inversion products available in AERONET collection, key for the analysis of aerosol absorption, the AERONET level 2.0 requirements reject inversion data if AOD values are less than 0.4, which dramatically reduces the amount of data. To address this issue, a similar procedure described by Mallet et al. (2013), Mateos et al. (2014) and Burgos et al. (2016) is applied. An

additional quality control is implemented to level 1.5 inversion data: the AOD must meet level 2.0 quality standards and all criteria regarding inversion products based on solar zenith angle and symmetry angles used in level 2.0 must be ensured. We have just removed the threshold regarding the AOD value, which is now set to 0.2 (see Dubovik et al., 2006; Mallet et al., 2013; Mateos et al., 2014). This consideration increases the uncertainty of the analyzed optical properties (refractive indices, single scattering albedo,…) but allows the characterization of a much larger amount of data. For instance, the SSA has an

absolute uncertainty of 0.03–0.07 depending on the aerosol load and type (Dubovik and King, 2000).

## 3 Time series and inventory of aerosol event days during A-LIFE experiment

The time series of the AOD at four wavelengths (380, 440, 1020 and 1640 nm) and the derived Ångström exponent (AE) are presented in Figure 1 for both CUT-TEPAK (Limassol) and Pafos AERONET sites. This figure shows only eight days that

can be considered as low turbidity conditions with AOD values below 0.2 for all channels (42.5% of total instantaneous data). Most of the days present, however, a notable impact of aerosol episodes of different kind. The AOD values (at 440 nm) span up to 0.4-0.8 on different days associated with two different AE categories: AE < 0.5 and 1.0 < AE < 1.5. The maximum value of AOD at 440 nm is obtained at the end of the experiment, on May 5th, 2017 when during three consecutive days high AOD values occur simultaneously with an increasing trend in AE values up to 1.5. These results illustrate the complexity of the area

with so many different conditions throughout the 1-month experiment.




The monthly median value of AOD at 440 nm for April is about 0.26 thus indicating the relevance of moderate-strong aerosol episodes during the whole experiment: 15.2% of total instantaneous data present an AOD larger than 0.4. Regarding AE values, 32% of the total instantaneous data present values less than 0.6 (coarse mode predominance) which can be attributed to coarse aerosols of mineral dust or marine type, according to typical classification values (e.g., Dubovik et al., 2002). Predominance
of fine particles is observed in 18% of total instantaneous data with AE > 1.4, meanwhile aerosol mixtures occur in almost half of the database with AE values between 0.6 and 1.4.

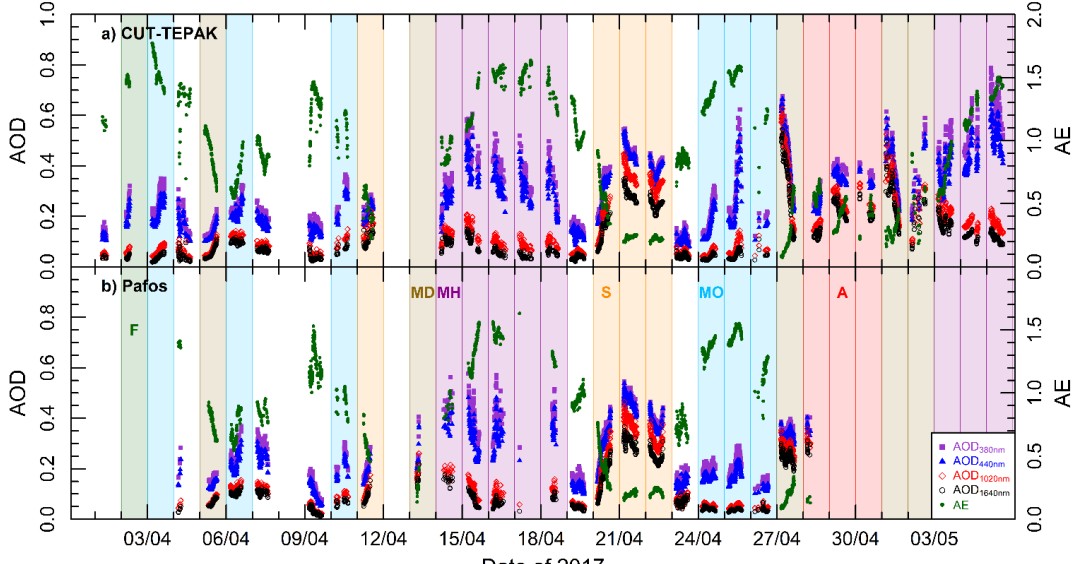

**Figure 1:** Columnar AOD (at 380, 440, 1020 and 1640 nm) and AE time series during A-LIFE experiment between 01/Apr/2017 and 05/May/2017 in two AERONET sites: a) Limassol (CUT-TEPAK) and b) Pafos. The background colors correspond to the inventory of aerosol event days described in Section 3 with the following flags: "F" event of fine particle, "MD" mixtures of Saharan and Arabian dust, "MH" mixtures of coarse and fine particles with high intensity, "S" Saharan dust event, "MO" mixtures of coarse and fine particles with moderate intensity, and "A" Arabian dust event.

A comprehensive inventory is carried out according to these columnar records. We have followed the same methodology that is described in detail by Cachorro et al. (2016) and Mateos et al. (2020), with meticulous human observation of all AERONET data throughout the experiment. All the days showing instantaneous values higher than 0.2 (this threshold is selected based on
the Percentile 85 method described by Mateos et al., 2020), are revised and AE values during that day are analyzed. We aim to classify all the columnar aerosol data into one of the following seven aerosol scenarios:
- Flag 'no': when no aerosol turbidity conditions are recorded
- Flag 'S': when **S**aharan dust aerosols are clearly identified
- Flag 'A': when **A**rabian dust aerosols are clearly identified
- Flag 'MD': when a **M**ixture of **D**ust layers occurs, and it is not possible to distinguish the dominant type
- Flag 'F': when the predominance of **F**ine particles is evident
- Flag 'MH': when there are aerosols of the fine and coarse modes simultaneously in the atmosphere with large values of AOD (predominant instantaneous data beyond 0.4 and AE > 0.5, **M**ixtures of **H**igh intensity)



- Flag 'MO': when there are aerosols of the fine and coarse modes simultaneously in the atmosphere with moderate values of AOD (most of instantaneous data are in the range 0.2-0.4 and AE > 0.5, Mixtures of **MO**derate intensity)

Each day is classified only as one of these aerosol types. In the event of a mixture or change in conditions, the day in question
is to be classified according to the most appropriate mixture category. Because of the complexity of the database, ancillary information has also been consulted (see Cachorro et al., 2016, for further details). In addition, we looked up aerosol profiles of the collocated LIDAR systems (Groß et al., 2024) to investigate the number of aerosol layers in the atmosphere. Air mass back-trajectories were also analyzed by means of the Hybrid Single Particle Lagrangian Integrated Trajectory Model (HYSPLIT, Stein et al., 2015) and "FLEXible PARTicle dispersion model" (FLEXPART; Stohl et al., 1998) calculations. The
addressed classification does not only consider the aerosol properties during the analyzed day, but the evolution during the previous or next days is also considered, in particular, if they present similar patterns. Some cases present certain extra-complexity, like April 27[th], 2017 when there are visible differences in the AOD evolution in CUT-TEPAK and Pafos sites, although the sites are only 40 km apart. The AOD at early morning presents large values with a low AE, thus indicating the presence of mineral dust aerosols. However, AOD evolution is quite stable at about 0.3 in Pafos and showing a notable
decreasing trend from 0.6 to 0.2 in CUT-TEPAK with an increase in AE up to 0.6. The final decision is made according to consulted ancillary information, and the whole day is classified as a mixture of dust.

Table 1 presents the inventory of all the aerosol event days during the whole experiment according to the columnar
measurements. As a result of the visual inspection, mineral dust occurred in 26 out of 35 days during the experiment, but under different conditions. A total of 12 days out of 35 presented only (or strongly predominant) mineral dust in the atmosphere being 4 cases originated in the Saharan desert, three days from Arabian desert, and five days presenting a mixture of these two types in greater o lesser extent. Fine particles were also abundant during the A-LIFE field experiment with 15 days out of 35, but mixtures (coarse and fine aerosol) is the most common situation: eight days present mixtures associated with high turbidity
(MH flag) and six days with moderate turbidity (MO flag). Note that this inventory is fundamentally based on columnar observations by solar photometry, and the classification could change if other kind of information is considered (e.g., in situ aircraft, LIDAR, etc). Groß et al. (2024) derived dust and non-dust backscatter and extinction coefficients from ground-based LIDAR measurements during A-LIFE experiment. Some days are equally identified by different techniques, such as April 21[st] 2017 as a Saharan dust plume over Cyprus. Records from April 5[th] 2017 are identified as Arabian dust by Groß et al. (2024),
but in the present study we cannot ensure only one dust type (microphysical aerosol properties are in this case closer to Arabian dust type, as it is discussed in Sections 3 and 4). It is worth mentioning here that the AERONET-based classification by Groß et al. (2024) only uses AOD and AE values from the photometer, meanwhile the present study introduces a more complex revision of all photometer data (including inversion derived parameters) as described above. Both inventories are in line, but some discrepancies are observed: the category attributed to April 11[th] 2017 in the present inventory is Saharan dust but mixed
dust is attributed according to AE values by Groß et al. (2024). The Arabian dust episode between 27-30 April 2017 shown in Table 1 also presents some days with different classification of mixed dust and polluted dust in the two studies.




**Table 1**. Inventory of aerosol event days obtained by sun-photometry during A-LIFE experiment. See Section 3 for legend of aerosol types with the following flags: "F" event of fine particle, "MD" mixtures of Saharan and Arabian dust, "MH" mixtures of coarse and fine particles with high intensity, "S" Saharan dust event, "MO" mixtures of coarse and fine particles with moderate intensity, "A" Arabian dust event, and "no" non turbidity conditions observed.

| April 2017 | | | | | | | | | | | | | | | | | |
|---|---|---|---|---|---|---|---|---|---|---|---|---|---|---|---|---|---|
| 1 | 2 | 3 | 4 | 5 | 6 | 7 | 8 | 9 | 10 | 11 | 12 | 13 | 14 | 15 | 16 | 17 | 18 |
| no | F | MO | no | MD | MO | no | no | no | MO | S | no | MD | MH | MH | MH | MH | MH |
| 19 | 20 | 21 | 22 | 23 | 24 | 25 | 26 | 27 | 28 | 29 | 30 | 1-may | 2 | 3 | 4 | 5 | |
| no | S | S | S | no | MO | MO | MO | MD | A | A | A | MD | MD | MH | MH | MH | |

## 4 Aerosol characterization during A-LIFE experiment by sun-photometry

### 4.1 Aerosol types in the AOD-AE space

Figure 2 displays the distribution in the AOD-AE space of all the instantaneous data classified according to the aerosol typing
in Table 1. The data of the three dust categories are located, as expected, in the bottom-right area. With a simple AOD-AE classification would have been impossible to distinguish between Arabian and Saharan dust, the low AE values obtained from fitting AOD between 440 and 870 nm do not show any difference. The 'MD' category presents the largest AOD values of dust categories. The whole area above dust is occupied by mixtures, since only one event is classified with 'F' flag and it is masked by the large number of points falling in the two mixtures categories ('MH' and 'MO'). Due to the small number of
measurements during that day (including direct and sky radiance data), this category is not analysed in further sections. As fine particles are present in mixture categories, they also play a key role during A-LIFE experiment. The 'MH' category is mostly placed in the upper-right side of the AOD-AE space, as can be also expected. Certain groups of points stand out in this space. For instance, the measurements on May 3[rd], 2017 classified as 'MH' category, are close to the dust cases in the AOD-AE space. The day before is classified as 'MD' because coarse mode is predominant and the days after are classified as 'MH'
since both fine and coarse modes are present in the atmosphere. Just on May 3[rd], 2017 there is a notable increasing trend in the AE values. The relevance of dust is evident on that day, but the presence of fine particles is also noticeable. This change of conditions proves 'MH' flag as the most adequate flag for that day, despite its AOD-AE position. The position in the space of the measurements on April 18[th], 2017 also classified as 'MH' seems to be more adequate for low turbidity or even 'MO' category, since they mostly present AOD < 0.2 and AE about 1.0. However, the evolution of the previous days is clearly
identified as 'MH' category and on April 18[th], 2017 the event is becoming weaker, achieving April 19[th], 2017 no turbidity conditions. Therefore, the 'MH' flag for that day is justified as the final conditions of a 5-days event with mixture conditions and large AOD values up to 0.6 and AE values between 1 and 1.5 (see Figure 1 and Table 1).

Focusing on the classification of Saharan and Arabian dust, despite occupying similar positions in the AE–AOD space (see Figure 2), their temporal evolution shown in Figure 1 shows different behaviour in terms of the separation between AOD
values at 1020 and 1640 nm. On days classified as Saharan dust events, the difference in AOD between these two infrared channels is notably larger. In contrast, this difference significantly decreases when Arabian dust is present in the atmosphere. To quantify this effect, we introduce a new Ångström Exponent parameter, $AE_{NIR}$, calculated using only 1020 and 1640 nm wavelengths, following the Ångström law:

$$AE_{NIR} = \frac{\log \frac{AOD_{1020nm}}{AOD_{1640nm}}}{\log \frac{1.640}{1.020}} \quad (1)$$



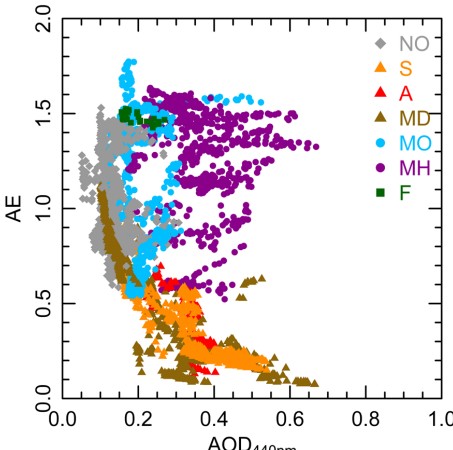

**Figure 2.** AE-AOD scatterplot for the seven aerosol types identified in Table 1. See Section 3 for legend of aerosol types with the following flags: "F" event of fine particle, "MD" mixtures of Saharan and Arabian dust, "MH" mixtures of coarse and fine
particles with high intensity, "S" Saharan dust event, "MO" mixtures of coarse and fine particles with moderate intensity, "A" Arabian dust event, and "no" non turbidity conditions observed.

This calculation is performed for all instantaneous data and classified by aerosol type, see Table 2. Results are presented only
for desert dust categories, since other types such as "MO", "MH" and "F" do not require this specific analysis. For comparison, the table also includes the typical Ångström Exponent (AE) values, which is the standard AERONET product, considering all channels between 440 and 870 nm. This typical AE shows high variability in the three analyzed aerosol types, with no statistically significant differences among them. In contrast, the proposed $AE_{NIR}$ parameter substantially reduces the dispersion. Saharan dust exhibits a higher $AE_{NIR}$ value, around 0.5. The value for Arabian dust category decreases up to 0.34, thus
indicating the presence of larger particles in the atmosphere during these events. This feature will be further examined in the next section through the analysis of the volume size distribution. The mixture category, "MD", displays an average $AE_{NIR}$ value very similar to that of pure Arabian dust, suggesting that dust mixtures during the A-LIFE experiment were likely dominated by Arabian sources. This effect will also be discussed in the following sections by the analysis of aerosol inversion products.


**Table 2**. Mean values (and Standard Deviation) of the Ångstrom exponent computed for the NIR range (1020 a 1640nm
channels, equation 1) and standard AERONET product (obtained fitting channels between 440 and 870nm), and number of data (N) falling in each type.

| Aerosol Type | $AE_{NIR}$ | $AE_{440-870}$ | N |
|---|---|---|---|
| S | 0.51 ± 0.06 | 0.34 ± 0.18 | 805 |
| A | 0.34 ± 0.06 | 0.44 ± 0.14 | 245 |
| MD | 0.33 ± 0.13 | 0.46 ± 0.32 | 624 |



### 4.2 Aerosol Microphysical Properties

Figure 3 shows the volume particle size distribution for the six aerosol types identified in the inventory shown in Table 1. We present the average of all size distributions for each type together with their standard deviation. Regarding dust aerosols, some discrepancies are observed when Saharan and Arabian mineral dust are compared. The shape of the curve for Saharan dust

peaks around a radius between 1 and 2 μm, meanwhile Arabian dust shows its maximum of the curve around radius of 4 μm. This can be a first indicator of the different properties shown by floating particles originated in these two deserts. The obtained effective radius of the coarse mode for Saharan dust aerosols is 1.55 ± 0.05 μm while the value is 2.12 ± 0.07 μm for Arabian cases. The volume mean radius (radius of the fitted lognormal distribution) of coarse mode is 1.85 ± 0.09 and 2.62 ± 0.11 μm for 'S' and 'A' cases, respectively, in concordance with the obtained effective radii.

The computed values for the 'MD' category are closer to the Arabian figures. For instance, the coarse mode effective radius for the dust mixtures is 1.95 ± 0.8 μm. Regarding mixture categories with fine and coarse aerosols, the coarse mode shape is like 'MD' curve and the fine mode peaks at small particles with radii about 0.1-0.2 μm, being the concentrations larger for the 'MH' category, as expected. Computed effective radii for fine mode are very similar: 0.16 ± 0.02 and 0.14 ± 0.03 μm for 'MH'

and 'MO' categories, respectively. With respect to the coarse mode, the values are placed in the middle between Saharan and Arabian values: 1.72 ± 0.19 and 1.87 ± 0.31 μm, respectively, showing higher dispersion because of the mixture conditions falling in these two categories.


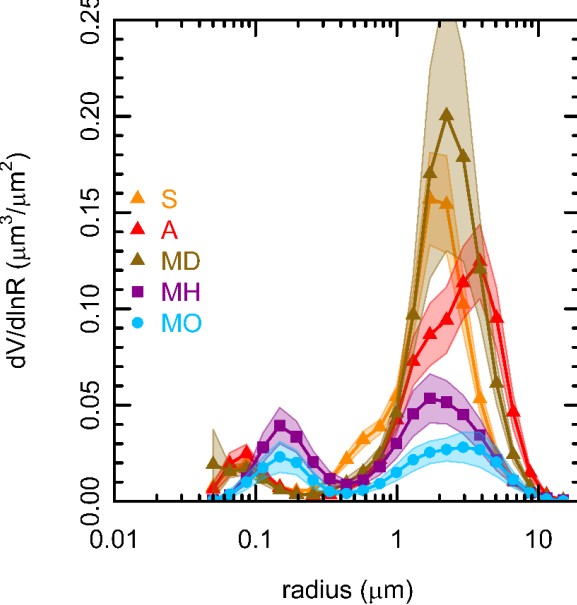

**Figure 3.** Volume particle size distribution for the aerosol types identified in the inventory of Table 1. See Section 3 for legend of aerosol types with the following flags: "MD" mixtures of Saharan and Arabian dust, "MH" mixtures of coarse and fine particles with high intensity, "S" Saharan dust event, "MO" mixtures of coarse and fine particles with moderate intensity, and

"A" Arabian dust event.



Figure 4 show the dependence of aerosol optical depth at 440 nm as a function of the columnar volume particle concentration (AOD vs $VC_T$), the latter being obtained as the integral of the volume particle size distribution. The relationship between these two quantities introduces the columnar volume efficiency factor ($E_V$) (e.g., Fraser et al., 1984; Kokhanovsky et al., 2009). This

quantity converts optical measurements into volume or mass concentrations if the aerosol type is well determined. Correlations between dust particle number concentration or dust volume concentration and dust extinction coefficient are also studied by Ansmann et al. (2019). Previous studies have reported empirical relationships between AOD and $VC_T$ (Prats et al., 2011; Toledano et al., 2012; Burgos et al., 2016). Burgos et al. (2016) analyzed $E_V$ for different categories of Saharan dust events in central Spain according to $VC_F/VC_T$ ratio (fine to total columnar volume particle concentration ratio). Their results indicate an

$E_V = 1.68$ $\mu m^2/\mu m^3$ for the coarse-mode-dominated cases ($VC_F/VC_T \leq 0.2$). As the fine mode gains weight, the value becomes larger up to $E_V = 3.7$ $\mu m^2/\mu m^3$ for $VC_F/VC_T \geq 0.45$.

Table 3 presents the results for A-LIFE experiment database, calculating $E_V$ at 440, 675, 870 and 1020 nm for the main five aerosol types. An additional category is included, representing a combined analysis of Arabian dust and mixed dust types. The maximum $E_V$ values are consistently observed at 440 nm across all aerosol types, with mixtures exhibiting the largest values.

$E_V$ decreases at 675 and 1020 nm, while a slight increase is noted at 870 nm for all aerosol types. The obtained results in the present study for 'S' category (when only Saharan dust particles are present in the atmosphere) match perfectly with the value of $E_V = 1.68$ $\mu m^2/\mu m^3$ reported by Burgos et al. (2016) despite the different locations (Spain vs Cyprus). The relationship between AOD and $VC_T$ is, therefore, an intensive property which can be used to identify aerosol types. All Saharan dust episodes shown in Table 1 are proved to be correctly identified. With respect the other reference value, an $E_V = 3.7$ $\mu m^2/\mu m^3$

is almost the limit for all the mixture categories, with experimental values close to 3 $\mu m^2/\mu m^3$ in this study. Most of the measurements conducted during the A-LIFE experiment lie between these two reference values described by Burgos et al. (2016) for dust mixtures.

With respect to Arabian mineral dust, the measurements of this type do not follow the same line than the Saharan dust. Arabian dust particles seem to present a smaller slope in the AOD vs $VC_T$ analysis. According to our data, an experimental fit of AOD

$= 1.37$ $VC_T$ ($R^2=0.985$) is found (see Table 3). But a larger amount of data, with a larger variability in AOD and $VC_T$ values, is still required to establish a more precise difference with the Saharan dust volume efficiency, but this figure can be a first approach to be used by further studies. The mixture of dust category presents points in between the two fit lines for Saharan and Arabian dust, and some extreme points with AOD > 0.6 seem to be close to Arabian dust, although a larger amount of data under different conditions is required, as discussed above. If we perform the linear fit with 'A' and 'MD' categories

together, the slope of the fit decreases down to 1.28 $\mu m^2/\mu m^3$ ($R^2=0.984$), which could be an indicator of a more real slope for Arabian dust.





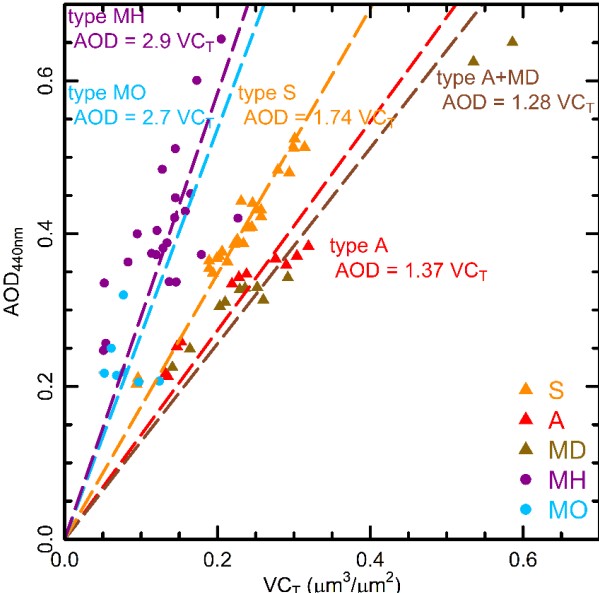

**Figure 4.** AOD (at 440 nm) vs $VC_T$ for the main five aerosol types identified in the inventory of Table 1. See Section 3 for legend of aerosol types with the following flags: "MD" mixtures of Saharan and Arabian dust, "MH" mixtures of coarse and fine particles with high intensity, "S" Saharan dust event, "MO" mixtures of coarse and fine particles with moderate intensity, and "A" Arabian dust event. Dotted lines indicate the empirical fits obtained in this study (see Table 3). The brown line indicates the linear fit for the volume efficiency factor of Arabian mineral dust and mixed dust categories analysed together.





**Table 3**. Values of columnar volume efficiency factor ($E_V$) obtained as the linear fit between AOD and $VC_T$ with a correlation coefficient $R^2$ using N number of data for four wavelengths. Values are obtained for the aerosol types described in Section 3 with the following flags: "MD" mixtures of Saharan and Arabian dust, "MH" mixtures of coarse and fine particles with high intensity, "S" Saharan dust event, "MO" mixtures of coarse and fine particles with moderate intensity, and "A" Arabian dust event. The "A + MD" category is also presented, with the joint analysis of these two categories.

| Type | $\lambda$ (nm) | $E_V$ ($\mu m^2/\mu m^3$) | $R^2$ | N | Type | $\lambda$ (nm) | $E_V$ ($\mu m^2/\mu m^3$) | $R^2$ | N |
|---|---|---|---|---|---|---|---|---|---|
| S | 440 | 1.737 ± 0.019 | 0.997 | 25 | MH | 440 | 2.919 ± 0.155 | 0.944 | 22 |
| | 675 | 1.500 ± 0.010 | 0.999 | 25 | | 675 | 1.424 ± 0.033 | 0.989 | 22 |
| | 870 | 1.582 ± 0.012 | 0.999 | 25 | | 870 | 1.769 ± 0.058 | 0.978 | 22 |
| | 1020 | 1.425 ± 0.010 | 0.999 | 25 | | 1020 | 1.276 ± 0.029 | 0.989 | 22 |
| A | 440 | 1.367 ± 0.053 | 0.985 | 11 | MO | 440 | 2.684 ± 0.401 | 0.882 | 7 |
| | 675 | 1.042 ± 0.012 | 0.999 | 11 | | 675 | 1.293 ± 0.088 | 0.973 | 7 |
| | 870 | 1.107 ± 0.020 | 0.997 | 11 | | 870 | 1.609 ± 0.154 | 0.948 | 7 |
| | 1020 | 1.011 ± 0.010 | 0.999 | 11 | | 1020 | 1.132 ± 0.075 | 0.974 | 7 |
| MD | 440 | 1.226 ± 0.044 | 0.987 | 11 | A+MD | 440 | 1.276 ± 0.036 | 0.984 | 22 |
| | 675 | 1.101 ± 0.015 | 0.998 | 11 | | 675 | 1.080 ± 0.012 | 0.998 | 22 |
| | 870 | 1.129 ± 0.020 | 0.997 | 11 | | 870 | 1.121 ± 0.014 | 0.997 | 22 |
| | 1020 | 1.077 ± 0.012 | 0.999 | 11 | | 1020 | 1.054 ± 0.010 | 0.998 | 22 |

### 4.3 Aerosol Absorption Properties

As the overall aim of the A-LIFE project is to deeply investigate the properties of absorbing aerosols focusing on mineral dust and black carbon mixtures, this section focuses on two main quantities regarding absorption power: the single scattering albedo (SSA) and the absorption Ångström exponent (AAE).

Figure 5 shows the Single Scattering Albedo (SSA) for the aerosol types observed during A-LIFE campaign. The shape of the SSA spectral dependency is similar for the three dust categories used in this study: SSA values increase from 0.90 at 440 nm to almost constant values for the rest wavelengths (670, 875 and 1020 nm). The various categories result in different values of the SSA in the longer wavelengths. Saharan dust exhibits larger SSA values around 0.99, which imply almost non-absorbing particles in the visible and near infrared ranges. This result is in line with previous studies in the area (Dubovik et al., 2002; Eck et al., 2010; Kim et al., 2011; Toledano et al., 2011; Giles et al., 2012, among others) but also at various western-central Mediterranean sites during desert dust outbreaks (e.g., Meloni et al., 2006; Cachorro et al., 2010; Valenzuela et al., 2012; Burgos et al., 2016). The decrease in SSA at shorter wavelengths is typical of desert dust absorption for aerosol with an iron mineral component (Sokolik and Toon, 1999; Di Biagio et al., 2019). Saharan dust particles sampled over Europe are dominated by $SiO_2$ and $Al_2O_3$, a characteristic they share with North American and Chinese dusts (Goudie and Middleton, 2001).

Arabian dust, however, exhibits maximum SSA values around 0.95 which imply a stronger light absorption power by this kind of dust. These values are even smaller than the previously reported for "Solar Village" site in AERONET (Dubovik et al., 2002). The 'MD' category is placed between the lines of the two pure categories.



With respect to the mixture categories, the typical spectral SSA behavior of urban/biomass burning aerosols is a sharp decrease with wavelength (see, e.g., Giles et al., 2012) and the result of the mixture is, therefore, a very soft spectral dependency, by compensation of the dependences of dust and fine particles. All the SSA values for 'MH' and 'MO' categories are almost constant, about 0.97 and 0.95, respectively. The absorption variability in mineral dust may be attributed to the variability of

iron-oxides contents in different dust areas (Kim et al., 2011).

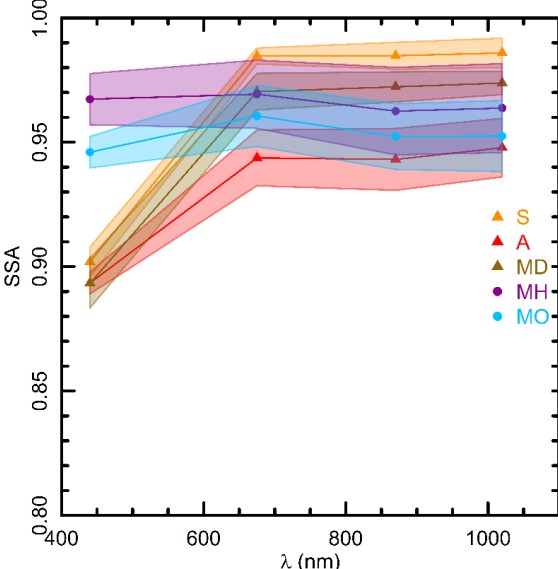

**Figure 5:** Single Scattering Albedo for the six aerosol types identified in the inventory of Table 1. See Section 3 for legend of aerosol types with the following flags: "MD" mixtures of Saharan and Arabian dust, "MH" mixtures of coarse and fine particles

with high intensity, "S" Saharan dust event, "MO" mixtures of coarse and fine particles with moderate intensity, and "A" Arabian dust event.

The absorption Ångström exponent (AAE) (see definition by e.g., Bond, 2001; Lewis et al., 2008; Russell et al., 2010; Mogo et al., 2017), describes the spectral dependence of absorption AOD, being dependent on particle size, shape, and chemical composition (e.g., Scarnato et al., 2013; Schuster et al., 2006; Li et al., 2016). An AAE value around 1.0 is usually attributed to Black Carbon rich aerosols (probably originated from fossil fuel burning), but this figure is modified to 1.05 and 0.90 for

fresh and aged black carbon particles, respectively (Liu et al., 2018). Larger values of AAE can be attributed to biomass burning or dust aerosols. Following Russell et al., (2010), mineral dust and their mixtures can present AAE values ranging between 2.27 and 2.34.




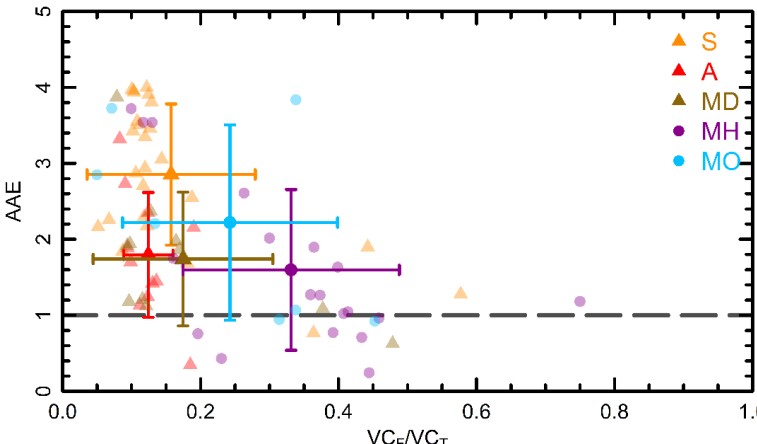

**Figure 6.** Absorption Ångström Exponent as a function of volume fine mode fraction for the six aerosol types identified in the inventory of Table 1. Transparent symbols point out all the inversion products available and solid symbols mean the average for this type with their corresponding error bars (±standard deviation). See Section 3 for legend of aerosol types with the following flags: "MD" mixtures of Saharan and Arabian dust, "MH" mixtures of coarse and fine particles with high intensity, "S" Saharan dust event, "MO" mixtures of coarse and fine particles with moderate intensity, and "A" Arabian dust event.

Figure 6 presents the values of AAE against the columnar fine mode fraction for each type of aerosols identified in Table 1. A mean AAE value around 1.6-1.8 is obtained for mixtures with stronger loads ('MH'), Arabian dust, and mixture of Saharan and Arabian dust. For mixtures of coarse-fine aerosol particles with weaker intensity the values increase up to AAE = 2.2 and the Saharan dust exhibits an AAE ≈ 3. For comparison, the AAE value is found to be dependent on dust contribution, ranging between 1 and 3 in two cities of the Iberian Peninsula in the period 2012-2015 (Fernández et al., 2017). Teri et al. (2025) presented a similar analysis using in situ aerosol data collected during the A-LIFE experiment. Cases dominated by purer dust exhibited higher AAE values, both at the surface and in the column-integrated measurements. Values for urban/industrial, biomass burning aerosols, and mixtures are previously reported between 1 < AAE < 2 (Giles et al., 2012). Typical values for dust (independently of origin) are about AAE = 1.9. So, the values for Arabian dust are in line with the previous ones, meanwhile Saharan dust exhibits an even larger figure in this study.

**5 Conclusions**

This study presents columnar aerosol data obtained by sun photometry during the "Absorbing aerosol layers in a changing climate: aging, lifetime and dynamics" (A-LIFE) field experiment, carried out in Cyprus island in April 2017. Products of AERONET network regarding aerosol load, microphysical and radiative properties have been analyzed. Following a meticulous human review of all columnar data and ancillary information detailed in section 3, the analysis provides an inventory of aerosol event days during the field experiment. Seven distinct aerosol types were identified: Saharan and Arabian dust as well as their mixtures, mixtures of fine and coarse aerosols with different intensity, events dominated by fine mode and non/low turbidity conditions. Mineral dust emerged as the predominant type throughout the experiment, being present on nearly 75% of the days. Calculations of the Ångström exponent in the near infrared (using AOD at 1020 and 1640 nm) range exhibit differences for Saharan and Arabian cases, serving as a useful tool for distinguishing between these two dust types.



The volume size distributions and optical properties obtained from the joint AERONET inversion of AOD and sky radiances have evinced different composition and size of mineral dust from Saharan and Arabian deserts. Larger particles, with more influence of aerosols with radii beyond 2 μm, are observed for Arabian cases. The analyzed mixtures show different behavior. Parameters from the volume size distribution for mixture of dust category, only coarse particles in the atmosphere, seem to

indicate that the obtained values are closer to the Arabian ones during "A-LIFE" experiment. When fine and coarse particles are present in the atmosphere, however, more variability is observed in the results in terms of predominant size and absorption properties.

One of the novelties of the present study is the calculation of the columnar volume efficiency factor ($E_V$), obtained as the linear fit between AOD and $VC_T$. This relationship has been demonstrated to be one intensive property associated to the aerosol type.

Values of previous studies confirm an $E_V = 1.68$ μm$^2$/μm$^3$ (with AOD at 440nm) as the reference value for Saharan dust aerosols. This slope decreases, however, for the analyzed cases of Arabian dust. We provide here an estimation of $1.28 \pm 0.04$ μm$^2$/μm$^3$ (using AOD at 440nm) which needs to be verified by long-term studies for this type of aerosol dust (pure cases).

The well-known spectral dependence of single scattering albedo of mineral dust is obtained in this study: lower values at the shorter wavelength (440nm) when compared with the rest of the wavelengths (675, 870 and 1020 nm). The Arabian dust

exhibits lower values in the four wavelengths used in the inversion products by AERONET, thus indicating a stronger absorption power by this kind of dust. The mixture of dust category ('MD') is placed between the two curves of the Saharan and Arabian pure types. No spectral dependence is observed for the mixture categories with fine and coarse particles in the atmosphere, as the opposite spectral dependence of mineral dust and biomass burning aerosols cancel each other.

Finally, mean values of the absorption Ångström exponent (AAE) for Arabian dust are about $1.8 \pm 0.8$ meanwhile for Saharan

dust are $2.9 \pm 0.9$. The category showing the smaller AAE values is 'MH' (fine and coarse aerosol particles simultaneously in the atmosphere with large values of AOD) with a mean value of $1.6 \pm 1.1$. No values close to 1.0 are observed during "A-LIFE" experiment, typically attributed to black carbon rich aerosols.

**Data availability**

Sun photometer measurements are provided by https://aeronet.gsfc.nasa.gov in the sites of 'CUT-TEPAK' and 'Pafos' (AERONET, 2025).

**Author contributions**

BW coordinated the A-LIFE project. DM, CT and AC organized sun photometer measurements during A-LIFE. AN is in charge of CUT-TEPAK AERONET site. DM, MHG, RR, SHA prepared the aerosol inventory. RG, DGF and CHB prepared the extra-filtered database of aerosol inversions. SG supported collocated lidar measurements. VEC and AF organized team campaign and founding. DM, CT, RR and VEC wrote the manuscript. All authors discussed the data and findings. All authors

reviewed the manuscript.

**Competing interests**

Silke Gross is member of the editorial board of ACP journal.

**Disclaimer**

Publisher's note: Copernicus Publications remains neutral with regard to jurisdictional claims made in the text, published maps, institutional affiliations, or any other geographical representation in this paper. While Copernicus Publications makes every effort to include appropriate place names, the final responsibility lies with the authors.



**Acknowledgements**

The A-LIFE field experiment was mainly funded by an ERC Starting Grant (A-LIFE) with support from the Deutsches Zentrum für Luft- und Raumfahrt (DLR) and the University of Vienna. This work was supported by the Ministerio de Ciencia
e Innovacion (MICINN), with the grant no. PID2021-127588OB-I00 and is based on work from COST Action CA21119 HARMONIA. This work is part of the project TED2021-131211B-I00375 funded by MCIN/AEI/10.13039/501100011033 and European Union, "NextGenerationEU"/PRTR. The authors acknowledge the support of the Spanish Ministry for Science and Innovation to ACTRIS ERIC. This work was also supported as part of EUBURN-RISK (S2/2.4/F0327), an Interreg Sudoe Programme project co-funded by the European Union. This study supported by the 'EXCELSIOR': ERATOSTHENES:
EXcellence Research Centre for Earth Surveillance and Space-Based Monitoring of the Environment H2020 Widespread Teaming project (www.excelsior2020.eu). The 'EXCELSIOR' project has received funding from the European Union's Horizon 2020 research and innovation programme under Grant Agreement No 857510, from the Government of the Republic of Cyprus through the Directorate General for the European Programmes, Coordination and Development and the Cyprus University of Technology. The authors also acknowledge the ATARRI project funded by the European Union's Horizon
Europe Twinning Call (HORIZON-WIDERA-2023-ACCESS-02) under the grant agreement No 101160258.

**Financial support**

The A-LIFE project was funded by the European Research Council (ERC) under the European Union's Horizon 2020 research and innovation programme (grant agreement no. 640458, A-LIFE). The A-LIFE field experiment received further funding from the ESA project A-CARE (ESA contract no. 400012581018NLCTgp).

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
