# Peer review of "Saharan and Arabian dust optical properties registered by sun photometry during A-LIFE field experiment in Cyprus"

_EGUsphere, 2025_

## Author Comment (AC1)

**#REF1**

Minor comment:

1. Abstract line 21: add be 'can be served'

The sentence was changed accordingly.

2. 'mineral dust being predominant'. Give a number percentage to 'predominant'

The sentence was rephrased: "…with mineral dust being present in nearly 75% of the days"

3. Usually dust is coarse…how can it serve as a key point for the entire size distribution?

It serves as a key point because it determines the shape of nearly half of the Volume Particle Size Distribution (VPSD) curve. As different size-related features are presented for the two dust types, Saharan and Arabian dust exhibit distinct VPSD shapes.

4. This line is redundant " Saharan dust exhibited smaller and less absorbing particles than Arabian dust." Given you already mentioned their angstrom coefficient.

The reviewer is right, we have rephrased the sentence: "According to the columnar records, Saharan dust also exhibited less absorbing particles than Arabian dust.".

5. Introduction line 16: " Larger and less absorbing particles are expected in Asian dust. " previously you said Saharan dust are smaller and less absorbing. So shouldn't Asian dust be larger and more absorbing? (give references)

We have rephrased this section to avoid confusion between Asian and Arabian dust. The comparison made by Su and Toon (2011) was between Asian and Saharan dust. The Taklimakan and Gobi deserts are the major dust sources in Asia. The new sentence is: Su and Toon (2011) found differences in the shape of particle size distribution between Asian (Taklimakan and Gobi deserts) and African dust.

6. Section 3: " Predominance of fine particles is observed in 18% of total instantaneous data with AE > 1.4, meanwhile aerosol mixtures occur in almost half of the database with AE values between 0.6 and 1.4. " give reference

We have rephrased the sentence to improve the clarity: "Fine particles (AE > 1.4) predominate in 18% of the measurements, whereas aerosol mixtures (AE between 0.6 and 1.4) account for nearly half of the dataset."

7. Section 3 lie 19 "All the days showing instantaneous values higher than 0.2…" are you talking about AOD? At what wavelength? Why this specific wavelength

We have added information about this issue: AOD at 440nm. This is the most common channel typically used in sun-photometry studies, since it is the smallest wavelength in original version of CIMEL sun photometers.

8. Figure 1: what does different wavelength say? Is it just because the instrument measured at these many wavelengths or is there any rationale behind using all?

The instrument measures at nine channels. For clarity, we have simplified the figure and focused only on two pairs: shorter wavelengths (380–440 nm) and near-infrared channels (1020–1600 nm). The latter are used to derive $AE_{NIR}$.

9. Does Figure 2 have data from both the sites? Make it clear

We have added two sentences on this topic. Throughout the manuscript, data from CUT-TEPAK and Pafos are jointly used in the analysis of aerosol characterization. The sentences added at the beginning of Section 4.1 are: 'The data from the two sites presented in Figure 1 are analysed together. As aerosol properties will be discussed by type in the following sections, both sites are considered equivalent for this purpose.'

10. Can you please describe in methods section how you derived volume distribution used in Figure 3? How does the volume distribution of F look like? Include that

That description was provided by Dubovik et al. (2000). We have rephrased the sentence introducing aerosol inversion products in AERONET. The shape of the fine mode can be observed in the curves for MO and MH types.

In Section 2, we have rephrased a paragraph on this topic: "Sky radiances at four wavelengths (440, 675, 870, and 1020 nm), combined with AOD, are employed to retrieve a set of aerosol optical and microphysical properties via inversion methods (Dubovik et al., 2000; 2006). These properties include particle size distribution, complex refractive index, single scattering albedo (SSA), phase function, absorption AOD, among others ..."

11. Can you also propose some clustering techniques given your criteria to automatically classify the dust type instead of meticulous human review? And compare to your classification

The reviewer is correct: clustering or a more automated method is indeed required and even preferable to the more meticulous manual review technique. However, for this campaign, we opted to obtain a more reliable inventory that can serve as a benchmark for testing automatic procedures. Clustering techniques will be addressed in the review paper of the campaign, which is currently in preparation and will be submitted soon. In addition, the classification of aerosol types using only sun/lunar photometry is necessary to enable comparisons with other studies conducted in different regions. One of the most important results of this study is the verification of the columnar efficiency factor obtained for Saharan dust in Spain and Cyprus, making this variable a very useful proxy in aerosol studies.

---

## Author Comment (AC2)

**#REF2**

Comments and Technical Corrections:

**Line 21, page 1:** The term "near-infrared range" can be vague. Please specify the exact wavelengths used (e.g., 1020 nm and 1064 nm) for better reproducibility.

The reviewer is correct. We have rephrased the sentence as follows: 'Ångström exponent values obtained from the 1020 and 1640 nm channels (0.5 for Saharan dust and 0.34 for Arabian dust) can be served as a classification criterion.

**Lines 16-17, Page 2:** The statement "Larger and less absorbing particles are expected in Asian dust" seems to contradict the Abstract, which indicates less absorbing particles for African dust. Please clarify this apparent inconsistency.

We have rephrased this section to avoid confusion between Asian and Arabian dust. The comparison made by Su and Toon (2011) was between Asian and Saharan dust. The Taklimakan and Gobi deserts are the major dust sources in Asia. The new sentence is: Su and Toon (2011) found differences in the shape of particle size distribution between Asian (Taklimakan and Gobi deserts) and African dust.

**Lines 22-23, Page 2:** The discussion of aerosol-cloud interactions could include the "cloud invigoration effect". Aerosol-cloud interactions depend not only on chemical composition but also on atmospheric conditions such as convection, atmospheric instability, and water vapor availability. Please consider expanding this discussion.

We followed the reviewer suggestion and added a concise sentence summarizing this interaction: "In addition, the invigoration effect is associated with changes in coupled cloud dynamics and microphysics caused by enhanced aerosol loading. This effect leads to diverse cloud responses that can vary depending on factors such as atmospheric instability, relative humidity profiles, wind shear, and the height of the freezing level (Altaratz et al., 2014)."

**Line 13, Page 3:** "several wavelengths," please specify all the exact wavelengths used by the CE318 sun-photometers in this study (e.g., 340, 380, 440, 500, 675, 870, 1020, 1640 nm).

We have added all the channels: "which performs direct sun measurements at 340, 380, 440, 500, 675, 870, 1020, and 1640 nm."

**Line 16, page 3:** Regarding "Ångström exponent (AE) by fitting AOD between 440 and 870 nm" - is the Ångström exponent estimated by fitting multiple wavelengths or calculated directly using the standard two-wavelength equation: $AE = -\ln(AOD_{440}/AOD_{870})/\ln(440/870)$? Please clarify the methodology.

The standard AERONET product at 440–870 nm is calculated by fitting multiple wavelengths. We have clarified this issue by adding: "…and the corresponding Ångström exponent (AE), obtained by fitting multiple wavelengths (in the 440-870nm range)".

**Line 23, Page 3:** Regarding "reject inversion data if AOD values are less than 0.4" - please specify if this threshold applies to a specific wavelength (e.g., AOD at 440 nm < 0.4) or to all spectral AOD values.

It is always referenced to 440 nm. We have added this information to the text.

**Line 25, Page 3:** Regarding "additional quality control is implemented to level 1.5 inversion data: the AOD must meet level 2.0 quality". Please clarify the quality control statement. Are you using level 1.5 inversion data with level 2.0 AOD as an additional filter, or should both inversion and AOD be at the same quality level (either both 1.5 or both 2.0)?

AOD is always required at Level 2.0. For inversion products, we use Level 1.5 data while applying the same filters as Level 2.0 inversion products, but without the strict condition of AOD > 0.4 at 440 nm. We have clarified the text accordingly: "The number of available AERONET Level 2.0 observations for SSA, which is a key parameter for analyzing aerosol absorption, is limited because SSA retrievals are considered reliable only when AOD at 440 nm exceeds 0.4, thereby substantially reducing the dataset. To address this issue, we applied a procedure similar to that described by Mallet et al. (2013), Mateos et al. (2014), and Burgos et al. (2016). The AOD data must always meet the level 2.0 quality standards. Level 1.5 inversion data are used only if all quality criteria for inversion products, based on solar zenith angle and symmetry angles, required for Level 2.0 are satisfied. We have just removed the threshold regarding the AOD at 440nm value, which is now set to 0.2 (see Dubovik et al., 2006; Mallet et al., 2013; Mateos et al., 2014)."

**Line 27, Page 3:** Regarding "removed the threshold regarding the AOD value". Please clarify: was the AOD threshold completely "removed" or was it "decreased" from 0.4 to 0.2? The sentence "removed the threshold" can be ambiguous.

Rephrased following the previous recommendation.

**Line 33, Page 3:** Regarding "Ångström exponent (AE)", please avoid repeating the definition as AE has already been declared in Line 16.

The reviewer suggestion is correct; we have avoided this repetition.

**Figure 1:** The contrast between AE and spectral AODs is insufficient for grayscale viewing. Please enhance the plot by using different line styles, markers, or increased line thickness to improve clarity.

We agree with the reviewer and have modified the style of Figure 1.

**Line 19, Page 4:** Regarding "showing instantaneous values higher than 0.2" - please specify what parameter this refers to. Is this spectral AOD at a specific wavelength? If so, which wavelength?

In this discussion, AOD is always referenced to 440 nm. We have clarified this point by adding this information.

**Lines 22 (Page 4) to 2 (Page 5):** Would it be possible to prepare a table summarizing the AOD and AE thresholds for each aerosol scenario? This would improve clarity and allow for more clear definition.

In this case, the identification of the aerosol scenario is not determined solely by AOD and AE values. Regarding mixtures, most of the data fall within the limits explained in the text; however, as shown in the AOD–AE space (Figure 2), in some cases the overlap of points makes it difficult to establish exact thresholds. For instance, purple points (MH category) appear in the typical zone of dust aerosols, while brownish points (MD category) are found over 'non-turbidity' cases.

**Lines 4 to 16, Page 5:** The classification approach, which includes not only aerosol properties but also synergy between models and the evolution of these properties, is very important. Would it be possible to prepare a flowchart or similar diagram to explain this methodology more clearly?

We appreciate the reviewer's suggestion regarding the inclusion of a flowchart to illustrate the classification methodology. In the current version of the manuscript, we have aimed to provide a detailed textual description of the methodology, highlighting the main sequential steps and their interconnections. While a diagram could indeed be a useful complement, the methodology itself is not new and is based on our previous works (see, e.g., Cachorro et al., 2016; Burgos et al., 2016). Moreover, in practice the classification can vary considerably depending on the day: sometimes it follows a relatively simple scheme, while in other cases it becomes highly complex, involving multiple interactions among different sources of information (aerosol properties and their evolution, aerosol profiles, models, air mass trajectories, etc.). For this reason, a single flowchart would not adequately capture the variability of the approach.

**Section 4.1:** What is the specific importance of aerosol classification using only sun-photometry? Does the classification in this section improve upon the classification presented in Section 3? Please explain more clearly the value and rationale for analyzing classification using only sun-photometry. I suggest including a brief introduction at the beginning of this section to explain why the classification of aerosol scenarios using solely sun-photometry is important.

Thank you for your comment. The classification using only sun-photometry is important because it allows comparisons with other studies in different regions where only photometric data are available. This approach ensures consistency and broad applicability of the results. While clustering or more automated methods are indeed preferable for large datasets, in this campaign we prioritized building a reliable inventory that can serve as a benchmark for testing such automatic procedures. These techniques will be addressed in the review paper of the campaign, which is currently in preparation and will be submitted soon. Additionally, one of the key outcomes of this study is the verification of the columnar efficiency factor for Saharan dust in Spain and Cyprus, making this parameter a valuable proxy in aerosol studies.

**Line 9 , Page 6:** Regarding "aerosol typing" do you mean "aerosol types"?

Yes, 'aerosol typing' is a common term to refer to the classification of aerosols into different types.

**Line 12, Page 6:** Please clarify the expression used for the Ångström exponent calculation. Consider using more precise wording such as "AE calculated from AOD at 440 and 870 nm" or "AE estimated from the wavelength pair 440-870 nm" instead of "AE values obtained from the fitting AOD between 440-870," which implies a multi-wavelength fit rather than a two-wavelength calculation.

To avoid misunderstanding, as mentioned in a previous suggestion, we have modified the presentation of the AE (obtained from AERONET) in Section 2.

**Line 32, Page 6:** Regarding "To quantify this effect, we introduce a new Ångström Exponent parameter, AENIR" - since the Ångström exponent has already been defined earlier in the text, it is not necessary to introduce it as a "new parameter." Instead, simply explain that AE is computed using the wavelength pair 1020-1640 nm and denote it as AENIR or AE(1020-1640) to distinguish it from AE calculated at other wavelength ranges.

The reviewer is correct; it is another calculation. We have rephrased the sentence as: "...we introduce a calculation of Ångström Exponent parameter, $AE_{NIR}$, calculated using only 1020 and 1640 nm wavelengths...".

**Equation 1:** This equation should either be presented earlier in the text when the Ångström exponent is first introduced, or it should be removed here and simply state that AENIR is estimated using the wavelength pair 1020-1640 nm (following the same methodology already defined for AE).

The AE product is obtained through multiple fitting (see previous comments), whereas AENIR is derived from a pair of wavelengths (Equation 1). Equation (1) is included at this point in the manuscript because it follows directly from Figure 1. Within the methodology, there is no need to add AENIR, as no different behavior is expected.

**Figure 2:** Please specify which wavelength pair the AE refers to (e.g., AE(440-870) or another pair of wavelengths).

To avoid misunderstanding, as mentioned in a previous suggestion, we have modified the presentation of the AE (obtained from AERONET) in Section 2.

**Line 11, Page 7:** Regarding "Ångström Exponent (AE)" - this parameter has already been defined in Line 16, Page 3. Please avoid repeating the definition. Consider checking throughout the text to remove redundant definitions of previously introduced parameters.

In the previous version, we introduced some definitions to remind the reader of key variables, but the reviewer is correct. We have removed the redundant definitions.

**Line 7, Page 8:** Regarding "coarse mode". Please explain how coarse and fine modes are defined. Are these based on a specific size threshold from the AERONET inversion algorithm, or should readers refer to Figure 3 for this information? Clarification of the size separation criterion would improve understanding.

Fine and coarse mode separation: The inversion code finds the minimum within the size interval from 0.439 to 0.992 µm. This minimum is used as a separation point between fine and coarse mode particles. It is emphasized that these component optical depths are defined optically (rather than in terms of a microphysical cutoff of the associated particle size distribution at some specific radius) and essentially depend on the fact that the coarse mode spectral variation is approximately neutral.

**Line 11, Page 8:** Please check the formatting consistency throughout the text for aerosol type abbreviations (e.g., "MD" vs 'MD'). Ensure the same format is used consistently for all aerosol categories.

The reviewer is correct; we have standardized the format throughout the manuscript.

**Line 11, Page 11:** How is the AAE (Absorption Ångström Exponent) estimated using AOD? Please clarify the methodology, as AOD represents total extinction (scattering + absorption), not absorption alone. Are you using AAOD (Absorption Aerosol Optical Depth) from AERONET inversions, or is there another method to isolate the absorption component?

AAE is obtained from the AERONET catalogue (we have now stated this more clearly in Section 2). It is based on the calculation of absorption AOD, defined as AOD [(1-SSA)*AOD_Extinction], which is derived using single scattering albedo data.

**Line 12, Page 11:** Regarding "Single Scattering Albedo (SSA)" - once defined for the first time, you do not need to repeat the full term. Simply use "SSA" in subsequent mentions.

We have removed the redundant definitions.

**Lines 16 to 18, Page 12:** Please consider moving this text to serve as the definition of AAE at the beginning of the section, rather than placing it at the end. This would improve the logical flow of the presentation.

The sentence appears at the beginning of the AAE discussion. Since the preceding part refers to SSA, we have kept the two discussions separate.